# Structural Basis of PE_PGRS Polymorphism, a Tool for Functional Modulation

**DOI:** 10.3390/biom13050812

**Published:** 2023-05-10

**Authors:** Eliza Kramarska, Flavio De Maio, Giovanni Delogu, Rita Berisio

**Affiliations:** 1Institute of Biostructures and Bioimaging, IBB, CNR, 80131 Naples, Italy; 2Dipartimento di Scienze di Laboratorio e Infettivologiche, Fondazione Policlinico Universitario “A. Gemelli”, IRCCS, 00168 Rome, Italy; 3Dipartimento di Scienze Biotecnologiche di Base, Cliniche Intensivologiche E Perioperatorie—Sezione di Microbiologia, Università Cattolica del Sacro Cuore, 00168 Rome, Italy; 4Laboratory Medicine, Mater Olbia Hospital, 07026 Olbia, Italy

**Keywords:** protein structure, tuberculosis, polymorphism, PE_PGRS

## Abstract

Background: The mycobacterial PE_PGRS protein family is present only in pathogenic strains of the genus mycobacterium, such as *Mtb* and members of the MTB complex, suggesting a likely important role of this family in pathogenesis. Their PGRS domains are highly polymorphic and have been suggested to cause antigenic variations and facilitate pathogen survival. The availability of AlphaFold2.0 offered us a unique opportunity to better understand structural and functional properties of these domains and a role of polymorphism in *Mtb* evolution and dissemination. Methods: We made extensive use of AlphaFold2.0 computations and coupled them with sequence distribution phylogenetic and frequency analyses, and antigenic predictions. Results: Modeling of several polymorphic forms of PE_PGRS33, the prototype of the PE_PGRS family and sequence analyses allowed us to predict the structural impact of mutations/deletions/insertions present in the most frequent variants. These analyses well correlate with the observed frequency and with the phenotypic features of the described variants. Conclusions: Here, we provide a thorough description of structural impacts of the observed polymorphism of PE_PGRS33 protein and we correlate predicted structures to the known fitness of strains containing specific variants. Finally, we also identify protein variants associated with bacterial evolution, showing sophisticated modifications likely endowed with a gain-of-function role during bacterial evolution.

## 1. Introduction

The mycobacterial PE/PPE/PE_PGRS protein family occupies approximately 10% of the coding capacity of the *Mycobacterium tuberculosis* (*Mtb*) genome [1,2]. The designation of PE and PPE family is due to the presence of Pro-Glu (PE) and Pro-Pro-Glu (PPE) motifs near the N-terminus of their gene products. PE and PPE families are further subdivided in subfamilies depending on their C-terminal domains [3,4]. The PE_PGRS subfamily includes the largest group of PE proteins and is present only in pathogenic strains of the genus mycobacterium, such as *Mtb* and members of the MTB complex, *M. marinum*, *M. ulcerans* and few other species, suggesting an important role for PE_PGRS proteins in mycobacterial pathogenesis [1,2].

PE_PGRSs show a peculiar modular structure, including an (i) N-terminal PE domain, highly homologous to the domain found in tens of others PE proteins, (ii) the polymorphic glycine-rich domain (PGRS), and (iii) a C-terminal domain, contained by a subset of PE_PGRS proteins that is unique for each protein and that varies in length from few tens to more than one thousand amino acids [5]. The PGRS domains contain multiple tandem repeats of GGA or GGN triplets that have been suggested to cause antigenic variations and aid in immune evasion mechanisms, thereby facilitating pathogen survival [1,6]. The fact that PE_PGRS proteins localise on the mycomembrane where they are properly exposed to interact with host components, lent support to this hypothesis [7,8] (see recent reviews for more details and conceptual model of cellular localization [5,9]). 

Due to the high content of Gly residues, PGRS domains have long been considered highly disordered since a high content of glycine residues is typically associated to highly flexible proteins [10]. Moreover, the alleged transition from a disordered to an ordered state has been associated to the ability of *Mtb* to hijack host immune machinery for subsequent pathogen survival [10]. Contrary to these presumptions, we have previously proposed a structural model for PGRS domains as poly-glycine II (PG_II_) sandwiches, in which Gly residues are not needed for the flexibility they may confer [5,9], but because they preferentially adopt a PG_II_ conformation [11] and are the sole residues to be sterically allowed. PG_II_ sandwiches are formed by left-handed antiparallel PG_II_ helices, sharing a similar conformation as poly-proline II [12,13], albeit stacked in two antiparallel groups, with glycine always pointing inwards, where no other residue would fit. More recently, the availability of AlphaFold2.0 offered a unique opportunity to better understand structural and functional properties of these sibylline domains [14]. Consistent with our previous models, AlphaFold2.0 predicts the structure of PGRS as PG_II_ sandwich domains of modular size, with a large variability in the number of constituting PG_II_ helices. Despite their size variability, PGRS domains all share similar structural features, that led us to describe them as “molecular sails” (Section 3.1). Indeed, like sails, they are all flat with a distinguishable straight edge made of short and regular loops rich in hydrophobic and aromatic residues (the sail foot). On the opposite side they expose loops of variable amino acid composition (the sail head).

PE_PGRS33 is among the most studied proteins of the family and is known to promote cell death and increase mycobacterial survival in macrophages, as demonstrated by heterologous expression in *M. smegmatis*, in a process mainly governed by its PGRS domain [7,15]. Its PE domain is responsible for protein translocation on the mycobacterial outer membrane so that the PGRS domain is exposed to the extracellular side [7,8], where it is available to interact with host components. It has been demonstrated that PE_PGRS33 plays immunomodulatory activities by directly interacting with Toll-like receptor 2 (TLR2), which triggers the secretion of several chemokines and cytokines, including TNFα, induces entry into macrophages and cell necrosis [16,17,18,19]. PE_PGRS33-mediated immunomodulatory properties were shown to depend on the PGRS domain [15,16,19]. 

To investigate the extent of PE_PGRS33 polymorphisms, genetic characterisation of the *pe_pgrs*33 gene on *Mtb* clinical isolates was performed, with the identification of sNSP (synonymous Single Nucleotide Polymorphisms, with no variation in protein sequence) and nSNP (non-synonymous Single Nucleotide Polymorphisms), insertions and deletions of one or more bp (indels) in the coding gene. The *pe_pgrs33* gene was shown to exhibit a low ratio of substitution rates at nonsynonymous versus synonymous sites (dN/dS ratio) but many in-frame indels, with 30% of the analysed strains containing large indels [17,20,21].

Interestingly, several strains harboured a frameshift indel and a premature stop codon, resulting in a protein reduced in length at the C terminus by ~30%. Although the impact of *pe_pgrs* polymorphisms on TB pathogenesis requires further inspection, it appears that large indels do not lead to a complete loss of protein function. Therefore, to better understand the effect of genetic diversity on the PE_PGRS protein family, we analysed the impact of most frequently observed polymorphisms of PE_PGRS33 on the protein structure and we seek to correlate structural changes with evolution and variant frequencies.

## 2. Materials and Methods

### 2.1. Selection of PGRS Sequence Study Set

Data on polymorphisms and indels occurred in the *pe_pgrs33* gene were obtained by a total of 1024 *Mtb* clinical isolates retrieved by six main studies as reported in Appendix A [16,17,20,22,23,24]. Phylogenetic information was available only for the *Mtb* clinical isolates reported in some studies [17,22,23]. Both polymorphisms and indels were annotated based on the nomenclature introduced by Talarico et al. [20] and reported in Appendix A with the indication of the variations in genomic position and aminoacidic sequence. Different alleles showing the same aa sequence were grouped based on the functional impact on the PE_PGRS33 protein and included in the same PGRS variant group. Frequency of each protein variant was then calculated and reported in Section 3.2. We related features on Rv1818c genomic variations (non-synonymous SNPs, deletion and insertions) to *Mtb* lineage by using characterised strains and corresponding information retrieved by McEvoy et al. and Camassa et al. [17,23].

### 2.2. Three-Dimensional Structures of PGRS Domains and Their Analysis

Three-dimensional structures of selected PGRS and C-terminal domains were computed using the Colab server (https://colab.research.google.com/github/sokrypton/ColabFold/blob/main/AlphaFold2.ipynb, accessed on 3 January 2023). This server predicts protein structures starting from their sequences using a slightly simplified version of AlphaFold v2.0 (AF) [25]. The reliability of the AF predictions was assessed both by the Local Distance Difference Test (LDDT) score and by the Predicted Aligned Error (PAE) score. The predicted pLDDT score (0–100) is a per-residue confidence score, with values greater than 90 indicating high confidence, and values below 50 indicating low confidence. PAE gives a distance error for every pair of residues, i.e., an estimate of position error at residue x when the predicted and true structures are aligned on residue y. Values range from 0 to 35 Å. If the relative position of two domains is confidently predicted, then the PAE values will be low (less than 5 Å). Structure superpositions were performed using the DALI software [26]. All structures were analysed and displayed using PyMOL [27].

### 2.3. T- and B-Cell Epitope Predictions

T-cell epitope prediction was performed using the software NetMHCIIpan-4.1, which allows predict binding to all human MHC class II isotypes [28]. The human MHC locus (in humans called HLA for human leukocyte antigens) is extremely polymorphic and encodes thousands of different HLA class II molecules, including HLA-DR, HLA-DP and HLA-DQ molecules. The method is based on artificial neural networks and has been trained on more than 50,000 quantitative peptide-binding measurements covering HLA-DR, HLA-DP and HLA-DQ, as well as two murine molecules. Strong binders (SB) were selected based on a %Rank score (lower than 1%), a transformation that normalises prediction scores across different MHC molecules. %Rank value of 1% means that a queried sequence obtained a prediction score that corresponds to the top 1% scores obtained from random natural peptides. Among best ranking epitopes we selected these with a predicted binding affinity to MHCII lower than 50 nM [28].

B-cell epitope prediction was performed using BepiPred-3.0 [29], a tool which utilises protein language models (LMs), trained on large datasets of protein sequences and structures. Best epitopes were mapped on the model structure using PyMol [27]. Moreover, structure-based B-cell epitope predictions were performed using PGRS33 AlphaFold2.0 model and the software ElliPro [30], which is trained on antibody–protein binding sites.

## 3. Results

### 3.1. Main Distinctive Features of PE_PGRS Structures 

Modelling of PE_PGRS proteins using AlphaFold2.0 provided us with key information to unveil the structural determinants of the peculiar features of these enigmatic proteins. Consistent with previous data, all modelled structures present a PE domain with an α-helical conformation, followed by the PGRS domain. Between the two domains, and conserved in the entire family, is the GRPLI motif (Figure 1) [5,14].

Focusing on PGRS domains, the predicted structure of the PGRS33 domain (or more precisely of the PE_PGRS33 variant expressed by the *Mtb* reference strain H37Rv, PGRS33^allRv^), is formed by 27 Gly-rich PG_II_ helices, stacked in two antiparallel groups. As previously described by us [14], a typical feature of PGRS domains is that they present a fully hydrophobic “sail foot”, which we proposed to be important for mycomembrane insertion. A single PG_II_ helix spans from three triplets at the N-terminal edge of the PGRS33 sail to a maximum of five triplets at its “head”. As shown in Figure 2, the PG_II_ sandwich is stabilised by hydrogen bonds involving backbone atoms of glycine residues (N—H-O and C—H-O). This feature and the high propensity of Gly to form PG_II_ conformations explains the high abundance of Gly residues in PGRS domains, as Gly is pointing inwards and is the sole residue to be sterically allowed (Figure 2). Moreover, there is an uneven distribution of hydrophobic and aromatic residues, which are mainly located at the sail “foot”, where they form an extended and compact hydrophobic surface (Figure 2A). These types of residues are also frequently localised on β turn loops (Figure 1) and sporadically on the side of PGRS, where they likely play a role in interactions with the host.

Inasmuch as modelled structures of PE_PGRS proteins were unavailable until the advent of AlphaFol2.0, the role of the GRPLI motif and the reasons for its strict sequence conservation were hitherto unknown. Analysis of the PE_PGRS33 structure provided an answer to this enduring question (Figure 2B). Indeed, the GRPLI motif plays an important role in stabilising the PGRS sail structure through hydrogen bonding and hydrophobic interactions. Specifically, (i) the full conservation of Gly114 of the GRPLI motif is likely to be ascribed to its torsional angles (φ = 62°, ψ = 5°), which are typical of Gly α_L_ conformation of the Ramachandran plot [31]; (ii) Arg115 side chain forms a hydrogen bonding interaction with the carbonyl oxygen of Gly135, and acts as a lock to stabilise the PG_II_ sandwich fold; (iii) Pro116 adopts backbone torsion angles in the typical antiparallel β-sheet region of the Ramachandran plot (φ = −70, ψ = 150), in a conformation restricted by the side chain pyrrolidine ring (the role of Pro116 may be to confer rigidity to the GRPLI motif, to offer a conformational lock); (iv) Leu117 side chain forms hydrophobic interactions with the side chain of Leu138 and takes part to the sail hydrophobic foot and (v) a similar hydrophobic interaction is observed between Ile118 and Ile139.

### 3.2. Structural Consequences of PGRS Polymorphisms

The sequence diversity of *pe_pgrs* genes of *Mtb* observed in several studies suggested that the corresponding surface exposed proteins might be responsible for polymorphisms and antigenic variability [1,6]. Sequencing of the *pe_pgrs*33 gene in *Mtb* clinical isolates confirmed the presence of tens of alleles, with different small and large in frame indels, SNPs and nsSNPs mainly occurring in the region coding the PGRS domain [17,23,24,32,33]. Despite these polymorphic regions, *pe_pgrs* genes, and *pe_pgrs*33 in particular, were found to have a dN/dS ratio < 1, suggesting that a purifying selection is acting on *pe_pgrs*33 [17,21]. However, the lack of knowledge on the structure of the PGRS domain and more specifically of the GGX repeats, has not allowed hypotheses of structural and functional consequences of these multiple indels. 

To address these issues, we have grouped all the *pe_pgrs*33 alleles sequenced in different studies using the nomenclature introduced by Talarico et al. [20] (Appendix A) and systematically collected all the available information on the *pe_pgrs*33 alleles obtained from more than 1000 *Mtb* clinical isolates, with the indication, when available, of the corresponding *Mtb* lineages for each allele (Appendix A). In this nomenclature, the different alleles and corresponding protein variants are compared to the *Mtb* H37Rv strain. We then grouped the alleles coding for identical PE_PGRS33 protein variants, so to assess the frequency and distribution of the proteins in the *Mtb* clinical isolates. As shown in Figure 3, most of the *Mtb* clinical isolates of the collection analysed express two types of PE_PGRS33 variants (PE_PGRS33^allRv^ and PE_PGRS33^all2^), that are associated with *Mtb* clinical strains of lineages 2 and 3. The few *Mtb* strains of the ancient lineage 1 show the PE_PGRS33^all26^ and PE_PGRS33^all29^ variants. It must be noted that since the collection of *Mtb* clinical isolates included in these tables cannot be considered representative of the *Mtb* strains circulating at global level, the frequency of each PE_PGRS33 variant can be used only as a proxy of each variant “success” or “fitness”.

Based on the relative frequency of each variant, their association with certain *Mtb* lineages and clades, and the feature of each polymorphism, we made extensive use of Alphafold2.0 to predict the structures of selected PGRS33 variants (Table 1), assuming the protein of *Mtb* H37Rv (PGRS^allRv^) as a reference [14]. To evaluate the reliability of models, we used both pLDDT and PAE scores (See Section 2). All computed structures are characterised by high reliability, apart from the connecting regions between PE and PGRS domains, which are predicted to be highly flexible. 

#### 3.2.1. Non-Synonymous Mutations (SNPs)

Sequence analyses of the PE_PGRS33 variants from the 1024 *Mtb* clinical isolates also identifies many SNPs in the PGRS33 domain, albeit with an overall low frequency (below 2%), and only one SNP in the GRPLI motif (L116P). Notably, in most of the cases, mutations of Gly correspond to the lowest frequency (0.10%), consistent with the key role of Gly residues in PG_II_ folds. Sporadically, a higher frequency (between 0.3% and 0.4%) is observed when Gly residues are replaced by small residues (Gly to Ser). Only in two cases, G210D and G388D, mutations to a bulkier residue are tolerated. However, these mutations do not occur in the PG_II_ body, but in β turn loops (Figure 4B), where a higher conformational freedom allows for accommodation of bulky residues. 

#### 3.2.2. The PGRS PG_II_ Sandwich Domain Can Tolerate Large Deletions and Insertions

A peculiar feature of the PG_II_ sandwich domains is their modular nature, with a large variability in the number of PG_II_ helices which compose PGRS sandwiches, since the hydrogen bonding network connecting PG_II_ helices (Figure 2) repeats symmetrically along the PG_II_ sandwich. Indeed, PGRS domains vary in size from the small sandwiches of PE_PGRS17, −18, −11 and −35 (7, 10, 11, 11 helices) to the medium sized as in PE_PGRS33 and −47 (27 and 29, respectively) to large ones as in Wag22 (54 helices). 

Consistently, two high frequency variants (Figure 3) contain large deletions, such as PE_PGRS33^all26^ (frequency 8.60%, I4; D14) and PGRS^all29^ (frequency 1,76%, I4; D2; D14; S13). Of note, deletion D14 results from a 1 bp deletion that causes an out of frame downstream the 337 aa residues and a “shorter” protein (374 aa) (see discussion). Structural analysis of these variants’ models supported the ability of the PG_II_ sandwich domain to preserve its structure despite the loss of large portions. Indeed, the deletion D14 in PE_PGRS33^all26^ determines the reduction in size of the PGRS33 sail to 15 PG_II_ helices, therefore large enough to form a stable PG_II_ sandwich domain (Figure 5A). Indeed, smaller PG_II_ sandwich domains exist in nature, with five or six antiparallel PG_II_ helices stacked in two antiparallel sheets, as in the case of the PG_II_ domain of the antifreeze protein sfAFP [34]. Moreover, the additional deletion of the region G139-L161 (D2) in PE_PGRS33^all29^, corresponding to two entire anti-parallel PG_II_ helices, leaves the structure and the distribution of hydrophobic residues unaltered, with the sole effect of further reducing the PG_II_ sandwich domain to 13 PG_II_ helices (Figure 5B). 

#### 3.2.3. Large Deletions Affect Immunomodulation 

A previous study of naturally occurring PE_PGRS33 gene sequence variations found that large insertions and deletions in the PE_PGRS33 gene significantly decreased the stimulation of TNF-α production, but small insertions and deletions as well as SNPs did not show a significant difference in stimulation of TNF-α production compared to the H37Rv-type PE_PGRS33 [16]. 

We analysed the predicted structural effects of the four immunomodulating deletions PE_PGRS33^all48^, PE_PGRS33^all49^, PE_PGRS33^all50^ and PE_PGRS33^all51^ [16]. Of these, PE_PGRS33^all48^ and PE_PGRS33^all51^ present the deletion of two PG_II_ helices (Figure 6). A larger deletion characterises PE_PGRS33^all50^, where the missing 91 residues form 6 PG_II_ helices (Figure 6). In all these cases, we observed that the effect of deletions did not alter the properties of the PGRS sandwich, albeit shortened of the deleted helices. Indeed, all inter-chain hydrogen bonds as well as the hydrophobic foot are well preserved. It is interesting to note that all deletions correspond to an even number of PG_II_ helices; this is the main feature that would allow PG_II_ sandwich full structure preservation upon deletion. 

A peculiar case is that of the PE_PGRS33^all49^. Indeed, different than in all other alleles, where an even number of PG_II_ helices were deleted, this allele is characterised by the deletion of three PG_II_ helices (Figure 7). This deletion is still predicted to preserve the PG_II_ fold. Indeed, it is worth noting that inter-chain hydrogen bonding interactions are predicted to involve only the protein backbone atoms, and therefore do not depend on the protein sequence. The deletion of entire GGX PG_II_ triples ensures the proper positioning of Gly residues along the PG_II_ helices. However, the conservation of the hydrophobic foot and the proper positioning of hydrophobic residues in the PG_II_ β-turn loops requires the deletion of an even number of helices. The predicted structural effect of a deletion of an odd number of helices produces is an upside-down inversion of the PG_II_ fold downstream the deleted portion. Consequently, the hydrophobic foot is positioned like in the PGRS33^allRv^ at the N-terminal region of the deleted G196-D243 region, whereas it is predicted to be located on the opposite side in the rest of the molecule (Figure 7).

All these findings suggest that the PGRS PG_II_ sandwich fold well tolerates even large deletions, without significant distortions. Indeed, even in the peculiar case of the PE_PGRS33^all49^, the modelled PGRS33 structure remains stabilised by all hydrogen bonds which characterise PG_II_ domains. However, while keeping the PG_II_ structural features, the deletion of PG_II_ helices in PG_II_ sandwich may result in weaker immunostimulatory activity (reduced TNF-α), possibly due to the involvement of the deleted portions in TLR-2 interaction [17]. Consistently, an anomalous concentration of hydrophobic residues, typically located on the PGRS “foot” or on β turns, characterise the central L237-G327 region. Nevertheless, the fact that these PGRS variants have been detected only in one *Mtb* isolate (frequency < 0.1%, see Table 1 and Figure 4A) indicates that they are among the less successful, likely due to the significant impairment of the PE_PGRS33 immunomodulatory properties. 

### 3.3. PE_PGRS33 Embeds T-Cell and B-Cell Epitopes

Epitope prediction was performed to identify those epitopes with high affinity of binding to MHCII, as the identification of peptides that bind to the MHCII molecules is of great importance for the design of new vaccines and immunotherapies. Using the software NetMHCIIpan-4.0, we identified several epitopes with high affinity to the MHCII complex, including HLA-DR, HLA-DP and HLA-DQ molecules [28]. Epitope prediction to detect epitopes recognised by MHCI, using the program NetMHCpan-4.1, produced no significant results. An overall number of 14 unique epitopes, binding to more HLA molecules, were found to display a binding affinity to MHCII molecules < 50 nM (Table 2). Of these, nine belonged to the PE domain and five to the PGRS domain of PE_PGRS33 (Table 2). Moreover, the PE domain contains epitopes with high binding affinity to the DRB1*0101 allele, which is the most common and widely distributed allele in human population [35,36]. This finding is consistent with a previous work, showing that the majority of predicted human T cell epitopes in PE_PGRS proteins are confined to the conserved PE domain [21].

PE_PGRS33 was previously observed to play a role in humoral response, with the PGRS domain as the main responsible of antibody generation. Indeed, anti-PE_PGRS33 serum was shown to be able to neutralise the proinflammatory activity of PE_PGRS33 in vitro [3]. Furthermore, mice immunised with native recombinant PE_PGRS33 were able to restrict *M. smegmatis* in vivo [3]. To understand which regions of the PGRS33 structure are involved in antibody recognition, we predicted B-cell epitope location using a two-fold approach (See Methods). B-cell epitope prediction using BepiPred3.0, identified no possible epitopes in the PE domain, whereas epitopes are predicted in correspondence to most exposed β turn loops between PG_II_-helices (Figure 8). Strongest epitopes are located at the C-terminal side of the PGRS33 domain (Figure 8). This finding was confirmed by structure-based epitope prediction with ElliPro, with the strongest epitope between residues 455–498 (Table 3).

## 4. Discussion

Structural characterisation of PE_PGRS proteins has been a major challenge due the peculiar structure of the GGX repeats that occupy the PGRS domain. We have recently proposed the organisation of these GGX repeats in tightly packed PG_II_ sandwiches [5,9], a structural organisation supported by the AlphaFold prediction software [14]. Many PE_PGRS proteins show extensive polymorphisms, mostly resulting from insertions and deletions in their respective genes. The classical and best studied example is the *pe_pgrs*33 gene, that has been sequenced in more than 1000 *Mtb* clinical isolates collected in different parts of the world. In this study, we aimed to investigate the structural and functional consequences of *pe_pgrs*33 polymorphisms on the predicted PGRS structure, also considering the frequency, and therefore the potential success, of the different allele/protein variants. Our experimental and methodological approach also provides a new view on the evolution of PE_PGRS33 in MTB complex. 

As we previously proposed, the organisation of the PGRS domains in PG_II_ sandwiches, consisting of tightly packed PG_II_ helices, provides a plastic structure that can tolerate large indels, while maintaining proper localisation of the unique amino acids that are found in the loops between the helices [9,14]. Interestingly, most PGRS variants detected with high frequency among the analysed *Mtb* clinical isolates present small insertions or deletions of three residues, thus leaving unaltered the predicted H-bonding patterns in the PG_II_ domain (as, for example I4, I6, I7, D5, D10 and D24). Although *Mtb* strains circulating at global level can vary, the data used for this study are gathered from multiple independent studies [16,17,20,22,23,24], and indicate an important trend among all PE_PGRS33 variants. For instance, the *Mtb* clinical isolates sequenced in Wang et al. [24] have been isolated in China and mainly include Lineage 2 *Mtb* strains. Most of these strains have the PGRS^allRv^ variant or the PGRS^all60^ variant, that contains the I6 insertion. These data well agree with experimental findings showing that small indels do not affect the immunomodulatory properties of PE_PGRS33 [16,17,22].

Instead, important determinants for structure preservation are (i) that inter-PG_II_ helices hydrogen bonding patterns are preserved, a requirement that does not depend on the sequence (since only main chain atoms are involved) and (ii) high preservation of the Gly residue of the GGX triplets, given the preference of Gly for the PG_II_ conformation and (iii) full preservation of the first Gly residue of the GGX triplets, since this Gly residue points inside the PG_II_ sandwich, where no other residue would fit. In addition to this, the preservation of the fully hydrophobic “sail foot” of the PGRS requires that an even number of PG_II_ helices are deleted or inserted. In this view, short insertions (or deletions) of GGX triples may be accommodated by prolonging (or shortening) the β turn loop close to the mutation. These structural observations are in line with the low frequencies of protein variants were these structural constrains have been violated due to nsSNP or indels (Figure 4A).

Machine-learning prediction algorithms such as AlphaFold2.0 can produce remarkably accurate protein models, albeit with possible regions of low confidence [25]. To be sure that all PGRS predicted structures presented high levels of confidence, we selected models with average pLDDT > 90 and per-residue pLDDT scores ranging between 70 and 100 [25]. These predicted structures suggest that large deletions, as in the case of the 91-residues deletion of PE_PGRS33^all50^, can be hosted by the PGRS scaffold with no main structure distortions. Indeed, our evidence is that structure preservation does not anti-correlate with the size of the inserted or deleted region. For large indels, we have observed that the deletion of an odd number of PG_II_ chains, albeit preserving hydrogen bonding interactions of the PG_II_ fold, inverts the sail “head” and “foot”, thus producing a distorted PGRS sail. This is the case of the allele PE_PGRS33^all49^, where the 49-residues deletion corresponded to 3 PG_II_ helices. Apart from the specific case of PE_PGRS33^all49^, which we predict to form a stable albeit distorted PGRS sail structure, we observed a full structural preservation of all deletion/insertion mutants. Nevertheless, despite their structural preservation, alleles embedding large deletions are usually associated to either a modified phenotype or to different immunomodulating properties [16,22] and, most importantly, with poor frequency (usually a single *Mtb* strain in the >1000 *Mtb* isolates analysed in this study). Taken together, these findings suggest that large insertion/deletion can be accommodated in the PGRS scaffold but can affect the immunomodulatory properties of PE_PGRS33, and these changes are associated with apparent lack of *Mtb* transmissibility. Consistently, large indels have been associated with non-transmissible forms of TB as non-cavitary TB [16,33] or with TB meningitidis in children [24]. Notably, these large polymorphisms seem not to affect virulence in an animal model of TB [17], albeit affecting transmissibility. Moreover, alleles with large deletions in the body of the PGRS33 PG_II_ sandwich (such as PE_PGRS33^all48^, PE_PGRS33^all49^, PE_PGRS33^all50^, PE_PGRS33^all51^) are associated to an attenuated elicitation of Tumour Necrosis Factor (TNF)-α release from macrophages, compared to the unmutated allele [16]. In all cases, PE_PGRS33 induced TNF-α release was proven to be TLR2-dependent. Therefore, it is likely that the deleted regions are involved in binding of PGRS33 to TLR2, in an event which is strictly connected both to macrophage entry and to the stimulation of the innate immune response, and can contribute to cavitation and transmissibility of the *Mtb* strain [17].

The analysis of all structural information obtained using AlphaFold2.0, combined to extensive analysis of sequences from *Mtb* clinical strains, suggested important clues on *Mtb* evolution. As shown in Figure 9, some relevant polymorphisms are associated with the ancient *Mtb* lineage L1 or with the L5 and L6 lineages of *M. africanum*, while other polymorphisms clusterise with the modern *Mtb* lineages (L1, L2 and L3). Most importantly, the ancestral forms of PE_PGRS33, PE_PGRS33^all26^ and PE_PGRS33^all29^ (Lineage L1) are significantly shorter than PE_PGRS33^allRv^, due to the presence of D2 and D14 deletions, with the latest embedding a frameshift mutation responsible for the shorter length of the PGRS domain. These observations depict a completely different scenario for the evolution and “fixation” of the different *pe_pgrs*33 alleles, suggesting that the ancestral *pe_pgrs*33 gene coded originally for a shorter protein, and following multiple insertions and deletions evolved in modern *Mtb* strains to become a gene coding a larger protein (498 aa for PGRS^allRv^). Hence, if we adopt an evolutionary view, what we defined as D2 should rather be considered an insertion and similarly what we defined I4 should rather be viewed as a deletion. The same will apply for other indels.

In addition to the structural impact of a short C-terminal side of PE_PGRS33^all29^ already described above (Figure 5), PE_PGRS33^all29^ also presents the non-synonymous SNPs S13 (see Appendix A for definitions). This SNPs, typical of lineage L1 (Figure 9A), results in the P116L mutation in the GRPLI motif at the N-terminal side of the PGRS domain, compared to the newer PE_PGRS33^allRv^ (lineages L2–L4). Hence, it transpires that the ancestral gene had a leucine at position 116 that was subsequently mutated in a proline. The comparison of PE_PGRS33^all29^ and PE_PGRS33^allRv^ modelled structures show no significant impact of this mutation on the backbone dihedral angles of Leu116 (Figure 10). However, the mutation L116P from the ancestral PE_PGRS33^all29^ to PE_PGRS33^allRv^ induces a conformational restraint imposed by the side chain pyrrolidine ring of Pro116. Likely, this mutation stabilises the protein since Pro116 confers rigidity to the GRPLI motif, thus offering a proper relative position of Arg115 to Gly135 for hydrogen bonding lock (Figure 2B). Indeed, all modern *Mtb* strains have a conserved Pro116. Therefore, if a time arrow is considered, evolution from the ancestral L1 lineage PE_PGRS33^all29^ to the L4 PE_PGRS33^allRv^ should be considered as producing three main changes: (i) the insertion in the PGRS33 domain of C-terminal residues 338–501, (ii) a 23-residues insertion (G139-L161) and (iii) the mutation L116P (Figure 10). Although all PE_PGRS33 variants are predicted to form a stable PG_II_ sandwich, it is likely that a larger PGRS domain in more recent lineages resulted, in a sort of a gain-of-function, in an improved exposure of surfaces able to interact with TLR2, thus mediating either the improved entrance into macrophages and/or more advantageous immunomodulatory properties that may have contributed to enhance *Mtb* transmissibility. Considering the presence of multiple T and B epitopes in the PGRS domain, we cannot rule out the involvement of the human adaptive immune responses in the selection and evolutionary process that shaped the evolution of PE_PGRS33 variants [17,21,37]. 

In conclusion, we provide structural basis for the observed polymorphism of PGRS33 domain. Given the limitations of computational approaches (e.g., Molecular dynamics) in the case of proteins interacting with membranes, especially when the exact membrane architecture is not known [38], we will validate our proposals using an experimental approach. Moreover, the functional and pathogenetic implications of the proposed model for the evolution of PE_PGRS33 require proper experimental testing in relevant models to fully appreciate their impact in the evolution of *Mtb*. Structural data cumulated here were precious to propose correlation between structure, stability, frequency, and evolution of PE_PGRS33. This analysis can be easily extended to other PE_PGRS proteins, provided that functional data are made available.

## Figures and Tables

**Figure 1 biomolecules-13-00812-f001:**
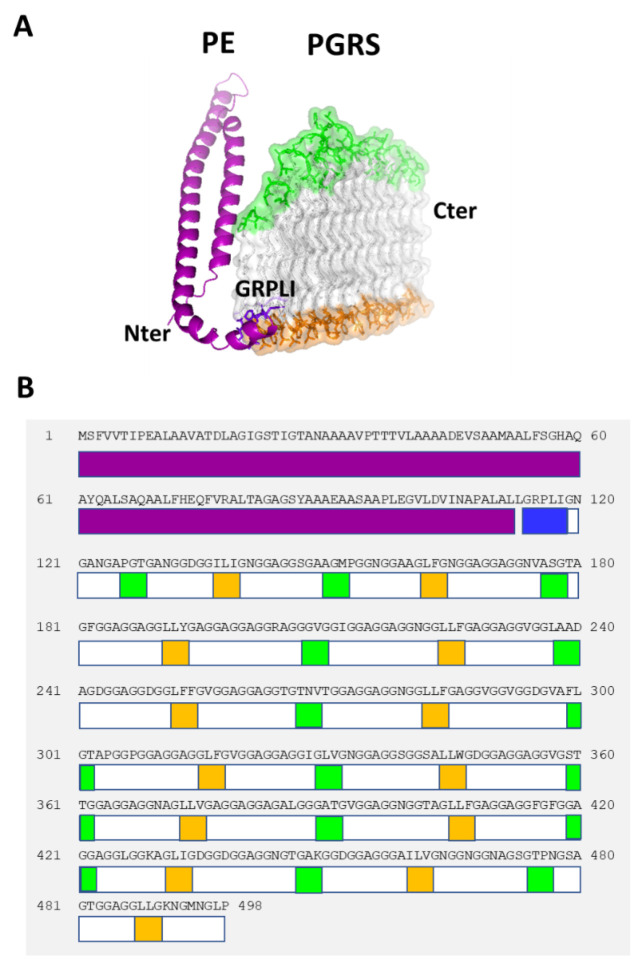
A schematic view of PE_PGRS33 structure. (**A**) Cartoon representation of PE_PGRS33 Alphafold2 model. (**B**) A scheme of domain boundaries and localisation of head and foot residues. The PE domain is represented in purple, the GRPLI motif in blue, head residues in green, foot residues in orange.

**Figure 2 biomolecules-13-00812-f002:**
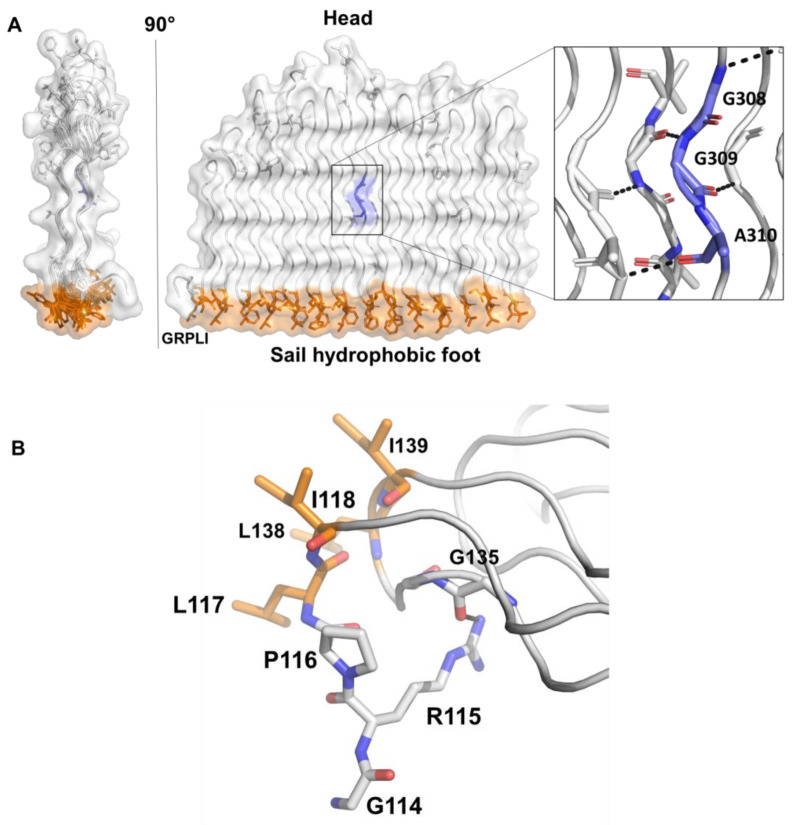
Structural features of PGRS33 domain and its GRPLI motif. (**A**) Surface and cartoon representations of PGRS33 domain (variant of the *Mtb* H37Rv reference strain). Hydrophobic residues are drawn in stick representation; those belonging to the hydrophobic foot in orange. A representative GGA triplet (308–310) is drawn in blue; the inset shows a detail of hydrogen bonding interactions formed by GGA triplets. (**B**) Stick representation of the GRPLI motif of PE_PGRS33 and of its interacting residues.

**Figure 3 biomolecules-13-00812-f003:**
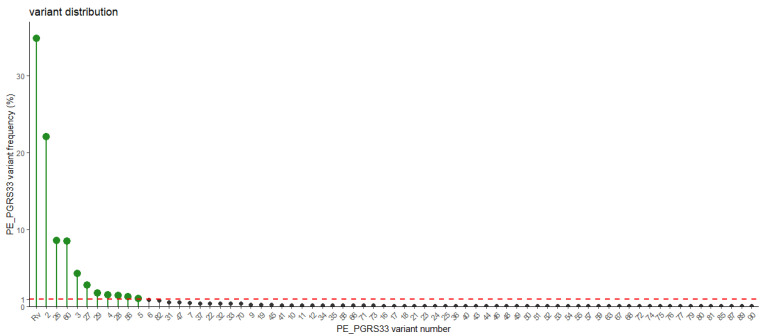
Frequency of the PE_PGRS33 variants as calculated in a collection of 1024 *Mtb* clinical isolates. PE_PGRS33 variants were functionally classified based on the impact of polymorphisms and indels on aminoacidic sequence. A red dotted line is drawn at 1% frequency.

**Figure 4 biomolecules-13-00812-f004:**
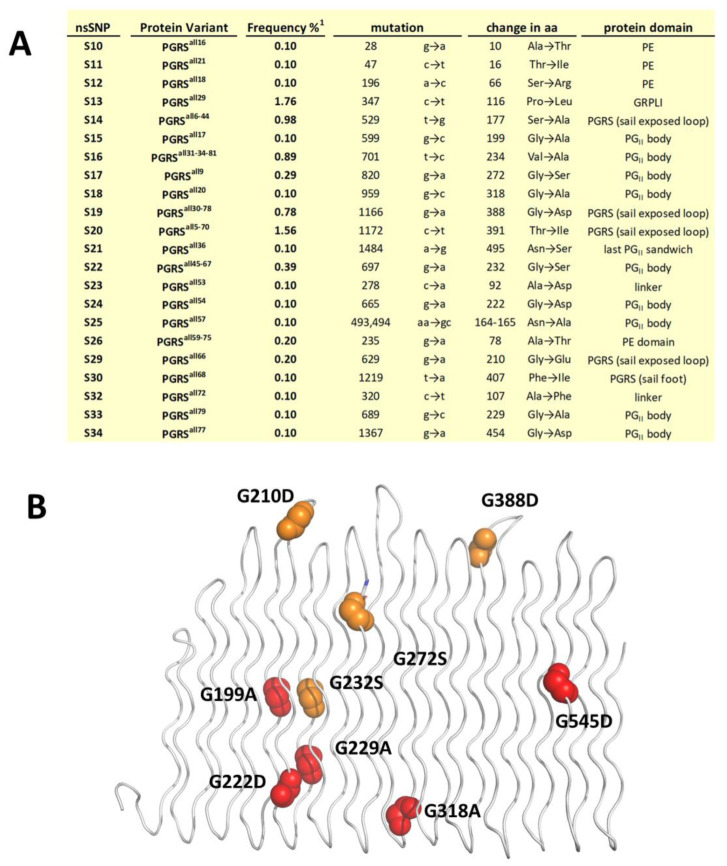
SNPs on PE_PGRS33 protein. (**A**) Frequencies of SNPs in PE_PGRS33 and (**B**) localisation of Gly mutations on the structure of PGRS33^allRv^. Gly mutations are shown as red (frequency 0.10) and orange (frequency > 0.10) balls. (Frequency%^1^ in panel A reports pe_pgrs33 allele frequencies as indicated in Appendix A and shown in Figure 3).

**Figure 5 biomolecules-13-00812-f005:**
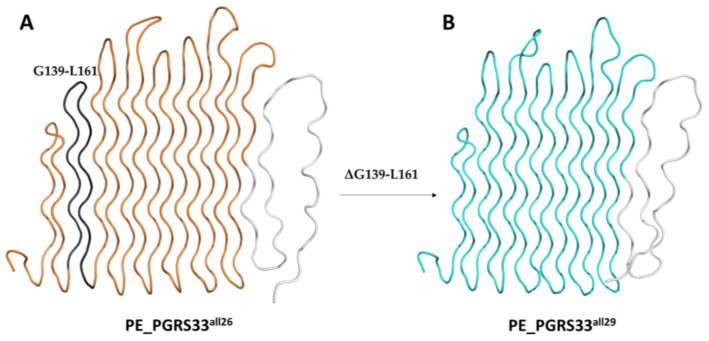
(**A**) Cartoon representation of PGRS33^all26^ (orange) and (**B**) PGRS33^all29^ (cyan). The region G139-L161 of PGRS33^all26^ (panel (**A**)) deleted in PGRS33^all29^, is drawn in dark grey. In both variants, a flexible region (white, pLDDT score < 50) originates from the D14 frameshift change. PE_PGRS33^all29^ also contains the nsSNP P116L, as reported below.

**Figure 6 biomolecules-13-00812-f006:**
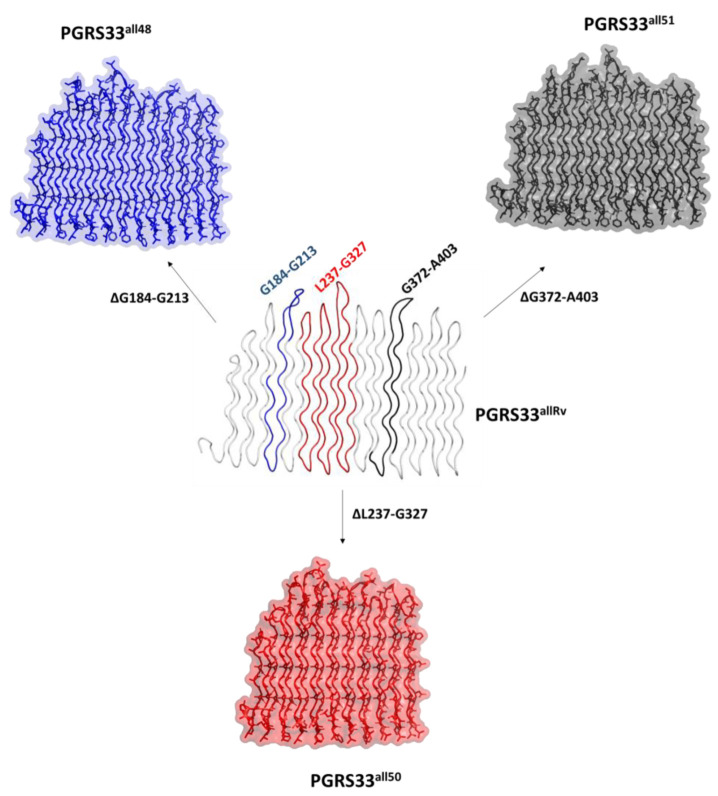
Cartoon representations of the effect of structure-preserving immunomodulating deletions on PGRS33 sandwich fold. Deleted regions are mapped on the wild type PGRS33 in the central panel.

**Figure 7 biomolecules-13-00812-f007:**
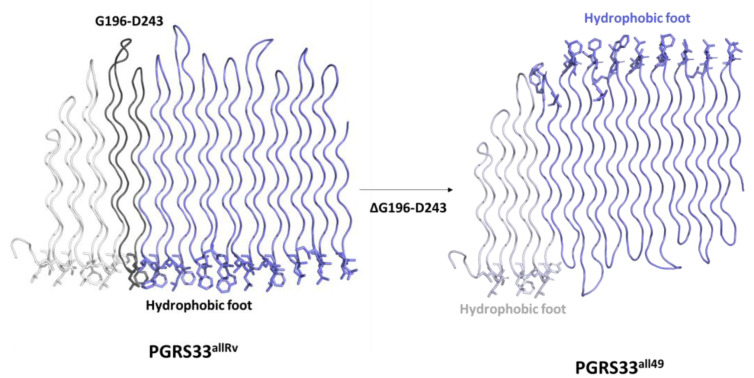
Cartoon representations showing the structural modification predicted by AlphaFold2.0 upon deletion of G196-D243. In the left panel, the three portions of PGRS33^allRv^ 114–195, 196–243 and 244–498 are drawn in grey, black and slate blue, respectively. The same colour code is kept in the right panel, displaying the structure of PGRS33^allΔG196-D243^. Hydrophobic residues belonging to the foot are drawn in stick representation.

**Figure 8 biomolecules-13-00812-f008:**
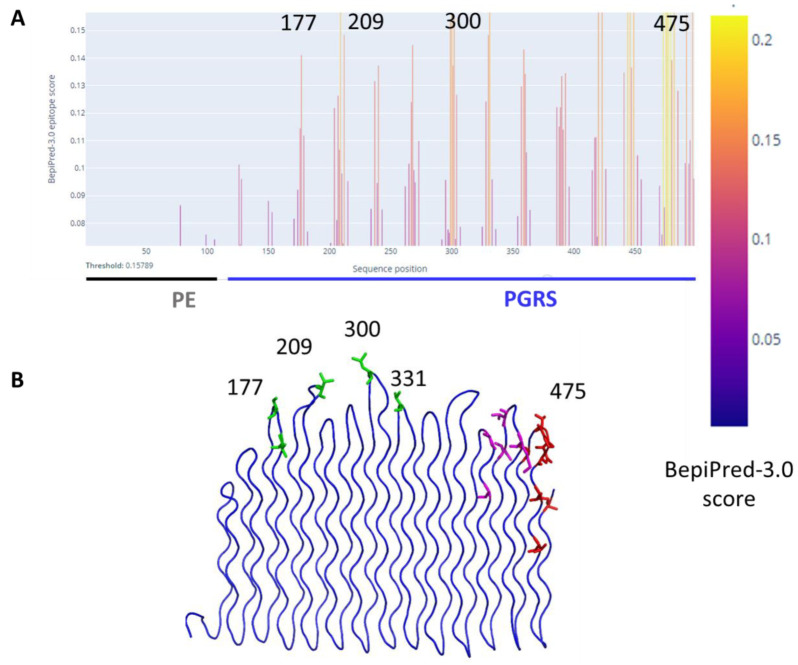
(**A**) Prediction of B-cell epitopes using BepiPred-3.0 and (**B**) mapping of most relevant epitopes (score > 0.15) on PGRS33 structure.

**Figure 9 biomolecules-13-00812-f009:**
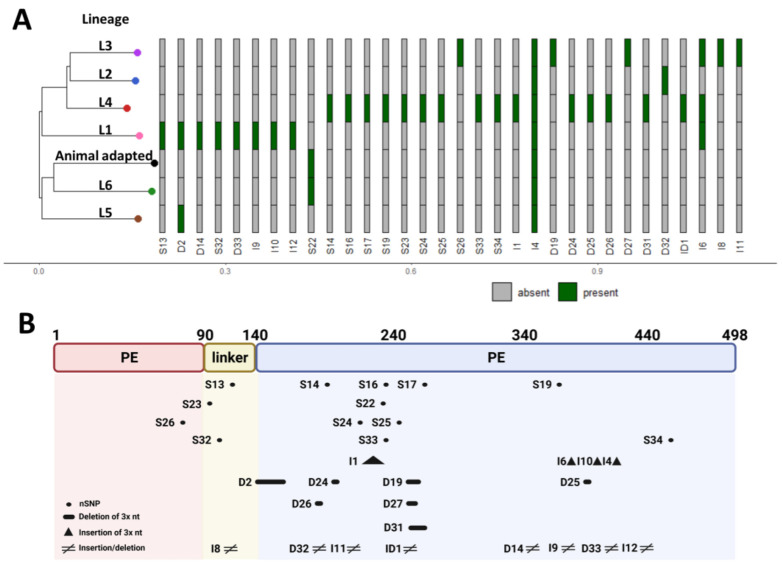
Polymorphisms occurring in *pe_pgrs*33 gene in *Mtb* complex lineages. (**A**) Presence of the most relevant *pe_pgrs*33 polymorphisms detected in *Mtb* and their association with the main *Mtb* complex phylogeographic lineages. (**B**) Graphical representation of the genomic changes in the *pe_pgrs*33 gene: S, non-synonymous mutations; D, deletions; I, insertions, according to the nomenclature as in [20].

**Figure 10 biomolecules-13-00812-f010:**
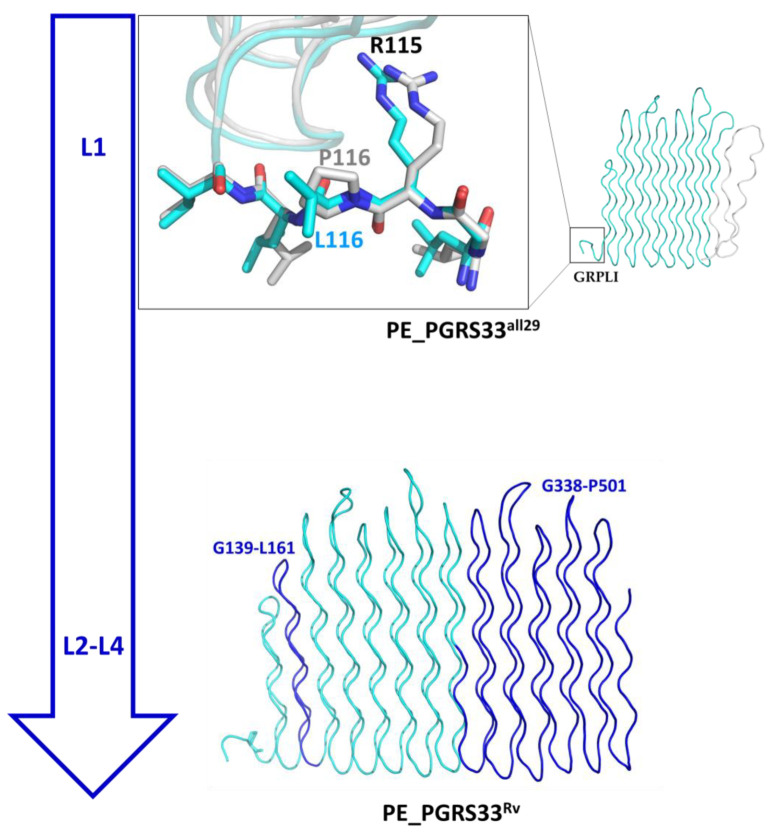
Structure evolution from the most frequent variant of lineage L1 PE_PGRS33^all29^ (**top**) to PE_PGRS33^AllRv^ (**bottom**). The inset in the top panel reports the superposition of GRLPI motifs of the two predicted structures; similar backbone torsion angles are observed for Pro116 and Leu116 (φ = −70, ψ = 150). In the bottom panel, inserted regions from PE_PGRS33^all29^ to PE_PGRS33^Rv^ variants are drawn in dark blue.

**Table 1 biomolecules-13-00812-t001:** Selected observed polymorphisms in PE_PGRS33. Alleles are numbered according to Talarico et al. [20].

PE_PGRS33 Variants	Lineage	Frequency (%)	Type of Protein Polymorphism	Reference
PGRS33^allRv^	L4	34.87		
PGRS33^all2^	L4	22.08	I4: 3-residues insertion (414-GGA-416)	[17]
PGRS33^all60^	L1-L3-L4	8.60	I6: 3-residues insertion (408-AGG-410)	[16,22,24]
PGRS33^all56^	L4	1.27	D24: 3-residues deletion (∆G214-A216)	[17,23]
PGRS33^all3^	L4	4.30	D10:3-resiudes deletion (∆A215-G217)	[20]
PGRS33^all29^	L1 EAI MAN	1.76	S13: Pro116Leu D2: 12-residues deletion (∆G139-L161) D14: STOP codon at position 374 I4: 3-residues insertion (414-GGA-416)	[17,20]
PGRS33^all26^	L1 EAI IND	8.60	D14: STOP codon at position 374 I4: 3-residues insertion (414-GGA-416)	
PGRS33^all45^	L5-L6 Animal	0.29	I4: 3-residues insertion (414-GGA-416) S22: Gly233Ser	[17]
PGRS33^all46^	L5-L6 AFRI	0.10	D2: 12-residues deletion (∆G139-L161)) I4: 3-residues insertion (414-GGA-416)	[16,20,33]
PGRS33^all48^		0.10	D20: 30-residues deletion (∆G184-G213)	[16]
PGRS33^all49^		0.10	D21 91-residues deletion (∆L237-G327)	[16]
PGRS33^all50^		0.10	D22: 32-residues deletion (∆G372-A403)	[16]
PGRS33^all51^		0.10	D23: 48-residues deletion (∆G196-D243)	[16]

**Table 2 biomolecules-13-00812-t002:** Predicted PE_PGRS33 T-cell epitopes with MHCII-epitope affinity < 50 nM.

	Epitope Position	Peptide	MHC	Affinity (nM)
**PE domain**	68	QAALFHEQFVRALTA	HLA-DPA10103-DPB12301	27.43
			HLA-DPA10103-DPB10201	16.01
			HLA-DPA10103-DPB10401	27.43
	59	AQAYQALSAQAALFH	HLA-DRB1_0101	4.85
			HLA-DRB1_0102	33.73
			HLA-DQA10501-DQB10301	39.33
	83	GAGSYAAAEAASAAP	HLA-DRB1_0101	16.17
			HLA-DQA10501-DQB10301	18.76
	73	HEQFVRALTAGAGSY	HLA-DRB1_0101	5.38
			HLA-DQA10501-DQB10301	49.65
	78	RALTAGAGSYAAAEA	HLA-DQA10102-DQB10602	50.02
	102	LDVINAPALALLGRP	HLA-DQA10102-DQB10602	23.61
	22	GSTIGTANAAAAVPT	HLA-DQA10505-DQB10301	14.46
	25	IGTANAAAAVPTTTV	HLA-DQA10501-DQB10301	10.71
	27	TANAAAAVPTTTVLA	HLA-DQA10505-DQB10301	12.32

**PGRS domain**	172	GGNVASGTAGFGGAG	HLA-DQA10501-DQB10301	37.44
	190	GLLYGAGGAGGAGGR	HLA-DQA10501-DQB10301	37.58
	281	GLLFGAGGVGGVGGD	HLA-DQA10501-DQB10301	30.13
	371	AGLLVGAGGAGGAGA	HLA-DQA10501-DQB10301	26.33
	452	DGGAGGGAILVGNGG	HLA-DQA10501-DQB10301	39.33


**Table 3 biomolecules-13-00812-t003:** Structure based predicted B-cell epitopes, according to ElliPro (score threshold 0.70).

Antigenic Region	Residues	Ellipro Score
AGGGAILVGNGGNGGNAGSGTPNGSAGTGGAGGLLGKNGMNGLP	455–498	0.77
LGRPLIGNGANGAPGTGANGGDGGILIGNGGAGGSGAAGMPG	113–154	0.77
GRAGGGVGGIG	203–213	0.74
GGNVASGTAG	172–181	0.71

## Data Availability

Data are available from the authors.

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
