# Peer review of "Structural Basis of PE_PGRS Polymorphism, a Tool for Functional Modulation"

_biomolecules, 2023, doi:10.3390/biom13050812_

Round 1

Reviewer 1 Report

The paper is describing only the studies based on homology modeling using the simplified version of AlphaFol. The AlphaFol server is not yet a brilliant tool especially when it comes to highly disordered proteins. It is not a tool to make predictions on mutations. For such study the MD simulation should be done. Another subject is that the structure predictions should be done by multiple tools designed for these specific tasks. If the proteins are highly disordered it is useful to perform folding using physical based methods.

Author Response

Comment

The paper is describing only the studies based on homology modelling using the simplified version of AlphaFol. The AlphaFol server is not yet a brilliant tool especially when it comes to highly disordered proteins. It is not a tool to make predictions on mutations. For such study the MD simulation should be done. Another subject is that the structure predictions should be done by multiple tools designed for these specific tasks. If the proteins are highly disordered it is useful to perform folding using physical based methods.

Response

We thank the referee for this comment, although our main goal is not to provide models but to correlate polymorphisms with structure and functional properties of these domains and investigate the role of polymorphisms in Mtb evolution and dissemination. By doing this, we agree with the referee that AlphaFold is not a good tool for highly disordered proteins and that if proteins are highly disordered it is useful to perform folding using physical based methods.

However, we propose that PGRS domains are not at all disordered, but forming the poly-glycine structure, stabilised by many intra-chain hydrogen bonds, in which Gly residues are not needed for the flexibility they may confer but because they preferentially adopt a PGII conformation and are the sole residues to be sterically allowed. This was our main point in our previous publication (Berisio and Delogu, Plos Pathogens 2022), and it is recalled in the Introduction of the submitted manuscript at lines 48-55:

“Due to the high content of Gly residues, PGRS domains have long been considered highly disordered since a high content of glycine residues is typically associated to highly flexible proteins [5]. Moreover, the alleged transition from a disordered to an ordered state has been associated to the ability of Mtb to hijack host immune machinery for subsequent pathogen survival [5]. Contrary to these presumptions, we have previously proposed a structural model for PGRS domains as poly-glycine II (PGII) sandwiches, in which Gly residues are not needed for the flexibility they may confer [3,6] but because they preferentially adopt a PGII conformation [7] and are the sole residues to be sterically allowed.”

We came to this proposal before Alphafold was invented (De Maio, Berisio et al. 2020), based on the homology of PGRS domains to the snow flea antifreeze protein, a compact and stable module composed of five tightly packed PGII helices (pdb code 2PNE). This compact module, also exists in the Salmonella phage S16 tail fiber adhesin (pdb code 6F45), demonstrating that even small poly-glycine structures (formed by solely five PGII helices) are stable. This is due to the strong contribution of intra-chain hydrogen bonds involving backbone atoms of glycine residues (N—H-O and C—H-O). This feature and the high propensity of Gly to form PGII conformations explains the high abundance of Gly residues in PGRS domains, as Gly is pointing inwards and it is the sole residue to be sterically allowed (Figure 1 in this manuscript).

Independent of flexibility, we agree with the referee that MD can be of extreme help where experimental data are lacking, and we use biophysical methods and MD simulations as a routine procedure (e.g. Squeglia et al. IJMS 2022, Ruggiero et al. Front Mol Biosci 2022, Moreira et al. IJBM 2020)). However, we have shown that PGRS domains embed characteristics of membrane proteins, as they present a fully hydrophobic “sail foot”, in agreement with studies demonstrating that the PGRS domain is available on the mycomembrane and can directly interact with host components, such as the TLR2 receptor (e.g. Palucci et al. 2016). This is a grey field in MD simulations, as it requests the exact knowledge of the interacting membrane, its exact composition, the proper choice of the forcefield and its parametrisation. Besides, computation time for MD these systems, which we have conducted in other studies where more knowledge of interacting membranes was available, are extremely long. On the other hand, it would not make sense to run MD simulations in water (e.g. TIP4) medium, which is the routinely used MD approach. For these reasons, we believe that an experimental validation (by recombinantly producing PGRS domains and characterising them using CD spectroscopy, calorimetry, and x-ray crystallography) will be sounder in this specific case. These studies, which are ongoing, will be the content of a future publication. To address the referee’s concern, we have included an extra sentence in the discussion section to concisely describe this concern (lines 534-537), which reads:

“Given the limitations of computational approaches (e.g. Molecular dynamics) in the case of proteins interacting with membranes, especially when the exact membrane architecture is not known, we will validate our proposals using an experimental approach. Also, the functional and pathogenetic implications of the proposed model for the evolution of PE_PGRS33 require proper experimental testing in relevant models to fully appreciate their impact in the evolution of Mtb.”

Reviewer 2 Report

I have reviewed the manuscript. It is a very interesting use of alphafold2 structure predictions to gain structure-function insights into a protein (using PE_PGRS33 as an example of this group of proteins) that is recalcitrant to other means of structure determination (x-ray for instance). And given the nature of this protein's involvement in Mtb infection, these understandings are important to further understand the virulence of the bacteria and look to strategies to combat them.  The authors explained their methods well, it was straightforward to follow, and their conclusions were supported by their observations.  There were a few typos (i.e. lines 142, 416 plus one or two others) that I am certain the authors will catch on revision.  I had not problems with accepting this manuscript given a few minor revisions.

Author Response

Comment

I have reviewed the manuscript. It is a very interesting use of alphafold2 structure predictions to gain structure-function insights into a protein (using PE_PGRS33 as an example of this group of proteins) that is recalcitrant to other means of structure determination (x-ray for instance). And given the nature of this protein's involvement in Mtb infection, these understandings are important to further understand the virulence of the bacteria and look to strategies to combat them.  The authors explained their methods well, it was straightforward to follow, and their conclusions were supported by their observations.  There were a few typos (i.e. lines 142, 416 plus one or two others) that I am certain the authors will catch on revision.  I had not problems with accepting this manuscript given a few minor revisions.

Response

We thank the referee for this strongly supporting comment. We have corrected the typos in the revised text (in track changes).

Reviewer 3 Report

The manuscript by Kramarska et al describes an interesting study of the PE_PGRS proteins, only present in pathogenic strains of Mycobacterium tuberculosis, by a combination of AlphaFold2 (AF2) structural predictions and phylogenetic analyses. The manuscript is interesting as provides pieces of evidence of the role of different polymorphisms and their connection with bacterial fitness.

While the methodology and conclusions are sound, an extra effort in translating the immunological terms and implications should be required to enlarge the scope for a broad audience. In this sense, I would suggest the following modifications/corrections:

-        Introduction Section.

o   Please first describe the PE/PPE/PE_PGRS protein family and the meaning of the different parts of that name. In other parts of the introduction section it is mentioned the “PE protein family” is that the same?

o   Is there any information about the location of these proteins on the mycobacterial cell wall? Please provide information about how these proteins anchor cell wall.

o   In the same sense, an overall scheme/figure showing the mycobacterial cell wall components and where PE_PGRS proteins are located should be added.

o   Line 78. Please explain the different terms: synonymous and nonsynonymous sites, in-frame indels…

o   Line 119. Please explain what HLA-DR, HLA-DP, and HLA-DQ molecules are.

o   Line 138. Maybe worth including a complete model for the whole protein with PE domain.

o   In the model showing the structural features of the PGRS33 proteins (Figure 1 and associated text), there is a great explanation about the sail hydrophobic foot and how the GCA triplets interact by an H-bond pattern. However, there is no explanation for the turns bending the backbone at the “head” regions of the protein. This is relevant as could have implications when describing the different polymorphisms. Could you explain it? Is there any common pattern?  It should be also interesting to provide a scheme showing how the full-length protein is split in subdomains based on the AF2 predictions (a very simplistic model showing how the full-length protein is divided into regions/subdomains separated by the sail hydrophobic foot on one side and by the residues composing the head in the other). It is possible to make a secondary-structure-like diagram for these proteins?  

o   Line 304. When describing how large deletions affect immunomodulation. Are these mutations affecting the exposed electrostatic potential? Concerning immunomodulation it is also important to know how exposed are these proteins (see my comments before).

o   In the discussion section. Please comment if PE_PGRS proteins are present in other bacterial species. Is that specific of Mycobacteria? This could be relevant to expand the impact of your analysis.

o    

Minor points:

-        Line 94 replace “table 2” by “Table 2”

-        Line 269 correct typo for “frequency”

-        Line327 correct typo for “helices”

Author Response

Comment

The manuscript by Kramarska et al describes an interesting study of the PE_PGRS proteins, only present in pathogenic strains of Mycobacterium tuberculosis, by a combination of AlphaFold2 (AF2) structural predictions and phylogenetic analyses. The manuscript is interesting as provides pieces of evidence of the role of different polymorphisms and their connection with bacterial fitness.

Response

We thank the referee for this strongly supporting comment.

Comment

While the methodology and conclusions are sound, an extra effort in translating the immunological terms and implications should be required to enlarge the scope for a broad audience. In this sense, I would suggest the following modifications/corrections:

-        Introduction Section.

o   Please first describe the PE/PPE/PE_PGRS protein family and the meaning of the different parts of that name. In other parts of the introduction section it is mentioned the “PE protein family” is that the same?

Response

The referee is right, as we gave it for granted. We have now provided an explanation to this naming at the beginning of the Introduction section (lines 36-43, track change).

Comment

Is there any information about the location of these proteins on the mycobacterial cell wall? Please provide information about how these proteins anchor cell wall.

Response

PE_PGRS were previously shown to localise on the mycomembrane. We have added a new paragraph with related literature at p. 3 (track changes), which reads:

The fact that PE_PGRS proteins localise on the mycomembrane where they are properly exposed to interact with host components, lent support to these hypothesis [7,8] (see recent reviews for more details and conceptual model of cellular localization [5,9]).

Comment

o   Line 78. Please explain the different terms: synonymous and nonsynonymous sites, in-frame indels…

Response

We have added this information to the main text (lines 90-94).

“To investigate the extent of PE_PGRS33 polymorphisms, genetic characterisation of the pe_pgrs33 gene on Mtb clinical isolates was performed, with the identification of sNSP (synonymous Single Nucleotide Polymorphisms, with no variation in protein sequence) and nSNP (non-synonymous Single Nucleotide Polymorphisms), insertions and deletions of one or more bp (indels) in the coding gene.”

Comment

o   Line 119. Please explain what HLA-DR, HLA-DP, and HLA-DQ molecules are.

Response

The referee is right. HLA stands for Human Leukocyte Antigen, whereas either DR, DP or DQ refer to the locus of their genes on the human chromosome. We have included HLA definitions in the abbreviations.

Comment

o   Line 138. Maybe worth including a complete model for the whole protein with PE domain.

o   In the model showing the structural features of the PGRS33 proteins (Figure 1 and associated text), there is a great explanation about the sail hydrophobic foot and how the GCA triplets interact by an H-bond pattern. However, there is no explanation for the turns bending the backbone at the “head” regions of the protein. This is relevant as could have implications when describing the different polymorphisms. Could you explain it? Is there any common pattern?  It should be also interesting to provide a scheme showing how the full-length protein is split in subdomains based on the AF2 predictions (a very simplistic model showing how the full-length protein is divided into regions/subdomains separated by the sail hydrophobic foot on one side and by the residues composing the head in the other). It is possible to make a secondary-structure-like diagram for these proteins? 

Response

We understand the concern by the referee, as this manuscript was focused on the PGRS domain. To remain focused on PGRS, we have included a new figure with the whole model in the supplementary material (Figure S1A). We also report, as requested, a simplistic picture showing how the full-length protein is divided into regions/subdomains (Figure S1B).

As shown in Figure S1, different than residues constituting the “foot”, sharing the characteristics of being hydrophobic, those of the “head” are highly variable (either hydrophobic or hydrophilic), consistent with their proposed role in interactions with the host.

Comment

o   Line 304. When describing how large deletions affect immunomodulation. Are these mutations affecting the exposed electrostatic potential? Concerning immunomodulation it is also important to know how exposed are these proteins (see my comments before).

Response

Deletions affecting immunomodulation described in the text include:

PGRS33all48, deletion 184-GAGGAGGLLYGAGGAGGAGGRAGGGVGGIG-213

PGRS33all49, Deletion 237-LAADAGDGGAGGDGGLFFGVGGAGGAGGTGTNVTGGAGGAGGNGGLLFGAGGVGGVGGDGVAFLGTAPGGPGGAGGAGGLFGVGGAGGAGG-327

PGRS33all50, deletion 372-GLLVGAGGAGGAGALGGGATGVGGAGGNGGTA-403

PGRS33all51, deletion 196-GGAGGAGGRAGGGVGGIGGAGGAGGNGGLLFGAGGAGGVGGLAADAGD-243

As it appears from the sequence, in all sequences except for PGRS33all50, there is at least one charged residue (PGRS33all48), 4 in PGRS33all49, 3 in PGRS33all51. Therefore, in most cases mutations affect the electrostatic potential. However, interactions with receptors are often also mediated by hydrophobic/aromatic interactions. As pointed out by the referee, exposure of these proteins needed for interactions with the host. As previously reported (Berisio, Delogu, Plos Pathogens 2022), we propose these proteins to be exposed to the host, through their anchorage to the membrane mediated by the hydrophobic foot.

Comment

o   In the discussion section. Please comment if PE_PGRS proteins are present in other bacterial species. Is that specific of Mycobacteria? This could be relevant to expand the impact of your analysis.

Response

PE_PGRS exist solely in mycobacteria. Also, as reported in the abstract, the mycobacterial PE_PGRS protein family is present only in pathogenic strains of the genus mycobacterium, such as Mtb and members of the MTB complex, suggesting a likely important role of this family in pathogenesis.

Comment

Minor points:

-        Line 94 replace “table 2” by “Table 2”

-        Line 269 correct typo for “frequency”

-        Line327 correct typo for “helices”

Response

We have corrected these typos.

Round 2

Reviewer 1 Report

The authors suggest that MD simulations are not applicable in this case due to the use of the TIP4 model of water. However, this can be changed, and other models can be used instead. Even hydrophobic parameters can be included in the simulations. The authors argue that MD simulations are too time-consuming to perform, but they could use graphical cards for acceleration, so it is not an excuse to avoid performing the simulations.

However, the authors' approach of modifying only one residue and using AlphaFold to predict its impact is insufficient. AlphaFold is not sensitive enough to detect this type of modification, and therefore, a more comprehensive approach is needed.

Author Response

We thank the referee for this comment. We are aware of the fact that Alphafold is not reliable for single-point mutations. Therefore, we never changed one residue but the smallest deletions/insertions we had considered was of three amino acid residues in most structured parts of the molecule (formed chapter 3.2.1). However, since this is still a grey area in assessing AlphaFold performance, we have now removed the specific section “3.2.1 Structural impact of small insertions and deletions”. We also changed the text and strengthened the concept that we present predicted structures.